# Frailty Syndrome—Fall Risk and Rehabilitation Management Aided by Virtual Reality (VR) Technology Solutions: A Narrative Review of the Current Literature

**DOI:** 10.3390/ijerph19052985

**Published:** 2022-03-03

**Authors:** Marek Zak, Tomasz Sikorski, Magdalena Wasik, Daniel Courteix, Frederic Dutheil, Waldemar Brola

**Affiliations:** 1Institute of Health Sciences, Collegium Medicum, The Jan Kochanowski University of Kielce, Zeromskiego 5, 25-369 Kielce, Poland; wbrola@wp.pl; 2Doctoral School, Collegium Medicum, The Jan Kochanowski University of Kielce, Zeromskiego 5, 25-369 Kielce, Poland; tomasz.sikorski@phd.ujk.edu.pl (T.S.); magdalena.wasik@phd.ujk.edu.pl (M.W.); 3Laboratory of the Metabolic Adaptations to Exercise under Physiological and Pathological Conditions (AME2P), Université Clermont Auvergne, 63000 Clermont-Ferrand, France; daniel.courteix@uca.fr; 4Occupational and Environmental Medicine, CHU, 63000 Clermont-Ferrand, France; fdutheil@chu-clermontferrand.fr; 5CNRS, LaPSCo, Physiological and Psychosocial Stress, Université Clermont Auvergne, 63000 Clermont-Ferrand, France

**Keywords:** frailty syndrome (FS), fall risk, rehabilitation strategies, virtual reality (VR) technology, telerehabilitation, exergaming, elderly, seniors

## Abstract

Frailty, a physiological syndrome (FS) affecting primarily the older adults, manifests itself through significantly depleted bodily reserves, and appreciably higher (up to over threefold) individual exposure to fall risk. Concomitant medical conditions such as balance impairment, reduced visual acuity, limited mobility, and significantly diminished daily functional performance further exacerbate the patients’ condition. Their resultant susceptibility to frequent hospitalisations makes their prognosis even worse. This narrative review aimed to provide an overview of published studies focused on rehabilitation management approaches aided by virtual reality (VR) technology in frail older adults. The authors had it also augmented with their own, evidence-based body of experience in rehabilitation. Making use of technologically advanced exercise machinery, specially adapted for rehabilitating frail older adults, combined with a structured exercise regimen, further aided by the application of select virtual reality (VR) technology solutions, clearly proved effective. Consequently, the patients were helped to move back from the frail to the pre-frail stage, as well as had their motor and cognitive functions appreciably enhanced. The application of modern technology in rehabilitating older adults over 65, affected by FS, when specifically aided by the select VR technology solutions, was also proven to complement successfully the conventional rehabilitation management. The overall versatility of the VR technology solutions, e.g., adaptation for home use allowing remote supervision, also makes this novel approach to rehabilitation far more appealing to the patients. They find it both very attractive and far more mentally engaging. Its considerable potential lies mostly in being appreciably more effective in bringing in desirable therapeutic outcomes.

## 1. Introduction

Fried et al. described frailty syndrome (FS) as a physiological syndrome, which, owing to the appreciably reduced capacity of respective physiological systems, tends to manifest itself through appreciably depleted bodily reserves and diminished resistance to stressors, to the overall detriment of the body as a whole.

FS may affect individuals of different ages, although it is most commonly encountered in older adults.

Given the anticipated rise in the proportion of seniors within the population, it is estimated that over 50% of individuals over the age of 85 may experience FS, also deemed one of the key factors contributing to disability in older adults. It is associated with an appreciable rise in overall fall risk, reduced self-reliance in the pursuit of routine activities of daily living (ADLs), frequent hospitalisation, and death [1,2,3].

In line with the definition proposed by the American Geriatrics Society, the hallmark manifestation of FS is supposed to be comprised of reduced resistance to stressors, combined with the dysfunction of the immune and endocrine systems. Presently, much attention is paid to the loss of cognitive function and the impact of various symptoms on a person’s physical, mental and social functioning, when addressing FS as a medical condition [4,5,6].

FS may be divided into three phases, depending on its severity. The first one is called the early phase (pre-frail) in which the symptoms directly indicative of the body’s limitations are encountered, even though an early response would ensure coping with this challenge successfully. The second one is the intermediate phase (frailty), characterised by hallmark symptoms indicative of limitations in individual functional capacity and deterioration of overall quality of life. The late phase (frailty complications) is diagnosed when individual functional self-reliance is significantly reduced, while the accompanying behavioural paradigm may potentially lead to death [7,8].

With a view to diagnosing potential FS patients, various clinical tests, some of them aided by dedicated instruments, are applied, in conjunction with specially designed, standardised questionnaires. The most frequently referenced criteria for defining FS are described in the phenotype model, and in the cumulative deficit model (CDM). By making use of the phenotype model, developed by Fried et al. in 2001, FS is diagnosed when at least three out of five listed components are ticked off. The CDM model, developed in Canada, initially comprised of 92 components, was eventually trimmed down to 36 (Table 1) [1,4,9,10].

### Objective

An overview of published studies focused on the FS issues in older adults, addressed in conjunction with overall fall risk assessment, and innovative therapeutic management options aided by virtual reality (VR) technology.

## 2. Frailty Syndrome

The causes of FS are not yet fully understood. The pathophysiological factors which affect the development of FS embrace abnormal metabolic processes, disorders of the endocrine and immune systems, coagulation disorders, musculoskeletal system, also both obesity and malnutrition [7,11].

Numerous studies corroborate the association between the development of FS and the elevated values of the variables accounting for the inflammatory process, e.g., C-reactive protein (C-reactive protein), factor VII, fibrinogen, D-dimers, interleukin 6 and 1β, tumour necrosis factor α and interferon gamma-induced protein 10 (IP-10) [12]. Those elevated values are associated with a drop in the level of such variables as IL-2, haemoglobin, insulin-like growth factor (IGF-1), albumin, and vitamin D in the blood [13,14].

The ongoing inflammatory process is also associated with oxidative stress, cellular ageing, and telomere shortening [7]. This adversely affects the cardiovascular, haematopoietic, and musculoskeletal systems, consequently contributing to hallmark phenotypic properties of the FS itself. One of the main manifestations of FS is sarcopenia, which is caused, inter alia, by reduced haemoglobin content in the bloodstream, owing to an increase in IL-6, characterised by a loss of body mass and a decrease in muscle strength. Furthermore, sarcopenia is often associated with osteoporosis and osteopenia [15].

The incidence of FS has also been linked to individual nutritional habits; low intake of protein and macro- and micronutrients in the daily diet being the acknowledged predisposing factors [16,17]. Reduced appetite in older adults may well lead to malnutrition and is generally associated with a loss of nutrients essential for the body functions [18]. In line with the recommendations of the European Society of Clinical Nutrition and Metabolism (ESPEN), older adults should consume 1.0 to 1.2 g protein/kg/d, whereas all malnourished seniors, or any individuals at risk of malnutrition, e.g., due to a specific medical condition, should consume 1.2 to 1.5 g protein/kg/d [19]. Both malnutrition and obesity, which are associated with elevated inflammation markers, increase the overall risk of FS [20,21].

Additionally, the individuals affected by type 2 diabetes (T2D) are at increased risk of frailty symptoms, owing to the appreciably reduced motor function [22,23].

FS is also associated with a low intake of nutrients such as vitamin A, E, B6, B12, D, and minerals such as zinc, selenium, and calcium [24,25]. Vitamin D and calcium are essential, especially in older individuals, as their deficiency may contribute to a loss of bone mass and muscle weakness, which, in turn, is associated with general bodily weakness and far greater exposure to fall risk and/or fractures [26,27]. Vitamin D deficiency occurs in approximately 90% of the older adults, mainly among the inhabitants of regions characterised by low exposure to sunlight [28,29]. The desirable concentration of 25-hydroxyvitamin D (25(OH)D) is not less than 30 ng/mL. Individuals at higher risk of FS have been established to have lower serum 25(OH)D concentrations [30].

Individuals affected by FS also remain at higher risk of endocrine disruption [11]. The hypothalamic–pituitary axis plays an essential role throughout the aging process and the onset of FS [31]. The association between the severity of FS symptoms and lower diurnal variation in cortisol levels had already been acknowledged [15]. It had also been established that a menopausal drop in oestrogen levels in women, and an age-induced drop in testosterone in seniors translates into an appreciable decrease in muscle mass and strength. This is accompanied by the decreasing levels of corticosteroids, dehydroepiandrosterone sulphate (DHEA), growth hormone, and insulin-like growth factor (IGF-1) [32,33]. Decreased IGF-1 levels bring about hormonal deregulation, associated with overall weakness and disability in older adults [34].

Jakubiak et al. also indicated in their literature review that the characteristics of frailty may well affect the clinical course of this condition, treatment efficacy, and complications of chronic lower limb ischaemia [35]. Moreover, the inflammatory process entailed in the pathogenesis of FS is also encountered in the pathogenesis of atherosclerotic cardiovascular disease and restenosis, following its endovascular treatment, as well as with regard to peripheral arteries [36].

Individual genetic predisposition may also affect the development of FS. Preliminary studies by Carini et al. highlight that the sTNF-R biomarker, which is strongly elevated in the individuals affected by FS indicating inflammation, is also accountable for FS. In the FS subjects, a decrease in miR-101-3p and miR-142-5p, as peripheral biomarkers and/or molecular effectors of frailty, are also encountered. Owing to certain limitations in the present study, further research is required, including a more numerous study group, blood samples duly divided into plasma and serum, so as to have these assumptions corroborated [37].

In older adults, FS stands for one of the major health issues. The Survey of Health, Aging and Retirement in Europe (SHARE) highlights that in the European community, more than 50% of adults happen to be at the first stage of FS or are pronounced as pre-frail [38]. FS is also found to be more common among women, individuals living on their own, low educated ones, those struggling with multiple medical conditions, and cognitive impairment [39,40]. The risk of FS is appreciably increased in the individuals who have experienced two or more hospitalisations within a single year, owing to an unexpected medical condition. A several-day-long hospitalisation is invariably associated with a decrease in muscle and bone mass [41].

There are several factors promoting the incidence of FS. They are better appreciated when stratified by the more or less developed countries (Table 2) [42,43,44].

The studies conducted to date readily furnish pertinent information on the actual prevalence of FS. On the other hand, the results of attendant analyses remain quite discrepant, owing mostly to a divergent selection of criteria and study groups [8]. In Europe, the average prevalence of FS in individuals over 65 years of age is 17%, with the lowest prevalence in Switzerland (5.8%), and the highest one in Spain (27%). When the individuals from the 50–64 years age range had been covered by the study, the lowest prevalence of FS was reported in Sweden (1.9%), and the highest one also in Spain (7.5%).

The prevalence of FS has been established as dependent on respective geographic regions, and the subjects’ age and gender. In the USA, the prevalence of FS ranges 7–12%, being appreciably higher in women [45,46]. By comparison, in the Polish study carried out in a group of adults over 60 years of age, FS was diagnosed in 40% of the subjects. This merely goes to show how ease of access to gerontologists within a particular country and the general availability of public healthcare services there may specifically affect the overall prevalence of FS [47].

When appraising oneself of the prevalence of FS in developing countries, it may easily be inferred that hunger, poverty, and limited access to healthcare services exert appreciable impacts on its onset and subsequent development. The study conducted by Wu et al. on the individuals aged 60 years and over established that in China, the country boasting the largest aging population in the world, 7% of Chinese are frail. The prevalence of FS varied, depending on the actual stratification by such variables as gender, age, education, and marital status. In the same study, FS was found more prevalent among the older adults, singles, economically disadvantaged, and women. Such individuals were also more likely to have sustained falls, and suffer from depression and assorted disabilities [48,49].

The studies conducted in Latin America established that FS was much more prevalent in those areas, compared with other world regions. The prevalence was 26.7% in Bridgetown, 39% in Havana, 39.5% in Mexico, and as high as 42.6% in Santiago de Chile [46].

## 3. Fall Risk

FS and falls are deemed truly resonant public health issues which affect older adults [50,51], whilst accounting for the second most common cause of injury-related deaths worldwide [52]. Overall, the risk of falls rises with age, acknowledged to be 1.16 to 3.6 times higher in the individuals already affected by FS [53]. In addition, some studies highlight this risk to be higher in the individuals who have already entered their pre-frail stage [54]. Fall risk in individuals from different FS-related groups is still being addressed by researchers worldwide. A more recent cohort study of 367 subjects established attendant fall risk to be the highest within the FS group, compared with the non-frail and pre-frail groups [55].

Falls may indeed result in injuries, a need for hospitalisation, subsequent limitation/reduction in one’s physical activity, resultant social isolation, and appreciably diminished overall quality of life. Consequently, FS-related falls pose appreciable challenges to all modern public healthcare systems [1,56,57].

Reduced muscle strength and body mass, high levels of fear of falling, and the resulting slowdown of gait speed, appreciably contribute to the development of FS, as well as to sustaining falls [58,59]. Fear of falling is associated with the impaired performance of dual-task activities, and a reduced ability to perform the activities of daily living (ADLs). Whilst making a direct reference to the current state of knowledge (2021), some studies tend to indicate that the prevalence of falls increases with age, with the frail individuals credited with a 59% higher prevalence of falls, as compared with their non-frail peers [60]. Falls are acknowledged to be causally combined with the natural changes occurring within one’s body, characteristic for the aging process, e.g., balance impairment, reduced visual acuity, mobility, and functional ability, which are merely further compounded by FS [61,62].

FS also interacts with an increased fall risk in the metabolic diseases, e.g., diabetes [63], hypertension [64], implying the need for introducing suitable modifications into the therapeutic management, including the actual number and specific type of medications regularly taken by the individuals affected by FS, with a view to reducing overall fall risk. Early diagnosis in the non-frail and pre-frail individuals is not a costly procedure [65] and should, therefore, be applied to ensure that an appropriate therapeutic regimen be selected for the individuals in whom FS has not yet developed, effectively minimising overall fall risk and an appreciable potential for hospitalisation.

The nature of symptoms encountered in the individuals affected by FS, especially in those living on their own, indicates the need for having them adequately trained, so they would be able to cope unassisted, having prior sustained an incidental fall. According to published research, 47% of older adults are unable to get up after a fall without having to rely on a third party’s help, even if it has not resulted in any serious injuries [66]. As far as the frail and pre-frail individuals are concerned, mastering the backward-chaining method (BCM) may well prove the most easily manageable solution for them [67,68].

Compared with the conventional method, the movements within the successive sequences are implemented starting off with the last position, i.e., standing upright. Once a specific movement is mastered by an individual, all previously learned steps are repeated until the last position is achieved, i.e., lying down, which is meant to reflect the most likely position following an incidental fall.

The first sequence of movements is made up of holding onto a chair and then bending the leg at the knee joint to touch the ground. The following steps entail moving on to kneeling on one knee, and then on both knees. The successive movement sequences should be split up into smaller parts in order to suit the individual capabilities of older adults. The individuals experiencing difficulties should be offered some extra assistance, e.g., by placing a pillow under their knees to reduce any undue discomfort. The final stage involves moving on to a supported kneeling position, and finally to a lying-down position [67,68,69].

When addressing the issue of overall fall risk assessment, it should be highlighted at this juncture that it is most often facilitated by way of carrying out a relevant scope of testing on the platforms fitted with an array of electronic sensors, including a stabilometric platform, in the AP axis (Y-axis, anterior–posterior (AP)) and ML (X-axis, medio–lateralis (ML)) planes. The tests executed on a diversity of platforms, aimed specifically at assessing any deflections of the body’s centre of gravity, are deemed objective and, therefore, regarded as the so-called gold standard, both with regard to the body’s centre of gravity shifts and overall fall risk evaluation. The actual sensitivity of detecting the fall risk has been proven at approx. 72% mark, while the subjects executed the Romberg test on a stabilometric platform. Inertial sensors analysing individual body movement paradigms are also widely acknowledged to have gained in popularity recently [70].

## 4. Application of Modern Physiotherapeutic Technology in the Individuals Affected by FS

Research on the effect of physical activity on improving the general health condition of the patients affected by FS over the last 25 years has yielded inconclusive and even contradictory results, when aiming to assess different variables with the aid of similar training programmes, inter alia, the effect of muscle strength training [71,72,73,74,75]. This has spawned the need for the application of modern technology solutions in addressing FS. Modern technology aids the diagnostic stage, overall prevention, and care through wireless camera sensors, Kinect™ sensors, smart homes incorporating environmental and physiological sensors, etc. [76]. The therapy/physiotherapy area has also borne witness to dynamic development, Nintendo^®^ Wii™, by far the most popular (37.5%) device in use [76].

As may easily be gleaned from a systematic review and network meta-analysis carried out by Marotta et al., even the relatively recently introduced device technologies, neither specifically devised nor intended for rehabilitation purposes, seem to hold substantial potential for being applied in therapeutic interventions in the older adults affected by FS [77]. Nintendo Wii is a commercial gaming device that uses a manual wireless controller and force platform. Xbox Kinect allows the user to interact with the game without a controller. Xbox Kinect does not provide visual references and space limits for sensors. The graphics environment of the Xbox Kinect is very detailed but less sensitive to detecting the displacement of the centre of pressure than the Wii Balance Board. The latter, as a platform, on the other hand, allows participation in the training program even for patients who have less integrity in cognitive performance and can be considered a visual reference [77].

The study by González-Bernal et al. (2021) [78] was conducted as a randomised trial involving 80 subjects (i.e., the study and control groups numbering 40 subjects each) in the individuals over 75 years of age, remaining in institutional care. The study group, unlike the control group, apart from conventional physiotherapy, also benefited from 20 rehabilitation sessions over 8 weeks, aided by virtual reality (VR), although without immersion in the Nintendo Wii VR technology. Following the study’s conclusion, the phenotype of bodily weakness altered for both groups. In the control group, this alteration was as follows: from 30 pre-frail and 10 frail subjects into 26 pre-frail and 14 frail ones. In contrast, in the study group, it altered from 23 pre-frail and 17 frail subjects into 34 pre-frail and 6 frail ones. Apart from a significant improvement in the bodily weakness phenotype in the study group, an increase in gait speed, an improvement in balance, and a drop in overall fall risk were also reported.

Furthermore, a study by Cicek et al. (2020) [79], carried out on 58 elderly subjects, focused on comparing the respective FS variables, demonstrated that both conventional physiotherapy, e.g., making use of a treadmill, an exercise regimen aided by a cycle ergometer, and some Nintendo Wii exercises, actually improved individual physical capabilities, as evidenced by the tests assessing functional ability and gait. In the Nintendo Wii group, however (8 weeks, 2 × per week for 30 min), this improvement was established to have been achieved to a greater extent, especially in the movement and balance variables. Jahouh et al. (2021) [80] additionally demonstrated that the therapy aided by Nintendo Wii reduced depression symptoms, anxiety, and apathy while enhancing individual memory and attention span, which translated into appreciably enhanced individual functionality in addressing the ADLs. Modern technology finds its application in the older adults and seniors affected by FS in view of its potential for providing specific guidance for the patients and their regular monitoring through remote control systems.

Individuals affected by FS often experience certain limitations associated with travelling to the therapy facility, which accounts for some training sessions to be missed by them altogether, when feeling weaker than usual. The application of remote physiotherapy technology through online sessions and microelectromechanical system sensors (MEMS) which record the actual movements of the person connected to them allows the therapists to track the effort exerted by the entire body through the sensors fitted inside the special bands to be installed on the patient’s joints. This facilitates the actual real-time monitoring of the progress, as well as controlling whether the pursued exercise regimen is being carried out in full compliance with the physiotherapist’s guidance [81].

In view of the ageing population and the resultant rise in the ageing index, society is faced with the challenge of putting into place effective organisational solutions aimed at promoting pertinent good aging policies locally for as long as practically manageable [82]. VR technology may well prove the right answer to such a quest, i.e., offering both a handy and easy-to-apply solution. Older individuals rate the exercises aided by VR technology as providing them with far more enjoyment than the traditional ones. It is also worth highlighting at this juncture that the exercises aided by VR also appreciably enhance both motor and cognitive functions [83].

Very few randomised control trials have been completed on the frail and pre-frail individuals, whilst carried out in the full immersion conditions. On the other hand, this has become the actual focus of the investigators who try to establish the optimal environment for the rehabilitation continuity, thus linking the hospital and the patient’s home [84,85]. Modern technologies are also versatile enough to be applied when modifying any previously developed training models, e.g., through replacing the traditional exercise machines with the ones specially refitted and adapted in a way more suitable for the individuals affected by FS.

In their study, Swales et al. (2021) [86] conducted a randomised controlled trial assessing the effect of strength training, aided by specialised machinery, on the functional capabilities of the individuals affected by FS. The study group consisted of 11 individuals affected by FS, aged over 65 years, who were trained for 6 weeks, 3 x per week, for 30–40 min each, on specialised machines featuring the built-in, smart-activated technology of starting the exercise at near-zero resistance, gradually increasing with the exercise time, which put them apart from the traditional machines used in the previous studies focused on assessing the same variables.

The results yielded by the study indicated a significant improvement in the subjects’ overall physical condition, who actually managed to have moved back from the frail to the pre-frail stage. The subjects’ gait speed increased by an average of 0.24 metres per second, and the short physical performance battery score by 1.50, i.e., through the improved lower limb muscle strength. The subjects also admitted to a significant improvement in their functional capabilities and self-confidence, when the training was still in progress.

Russo et al. [87] conducted a pilot study making use of BTS-NIRVANA (BTs-N) technology. NIRVANA is a device based on optoelectronic infrared sensors which allow the patient to interact with a virtual scenario. The system is coupled with a projector or large screen displaying a series of interactive exercises, whereas the infrared camera keeps tracking the patient’s movements. This prompted the investigators to come to a preliminary conclusion that VR training with BTs-N may well enhance individual cognitive abilities. Further research is obviously required to ensure that those findings are given sufficient credence, especially when conducted on a much more numerous study group.

Modern technology should, therefore, complement a conventional therapy, focused primarily on pain reduction, improvement of motor functions, flexibility/agility, muscle strength enhancement, aerobic capacity, dynamic balance, and postural control. This might effectively be achieved by introducing attractive variability to the training regimen, having it remotely supervised when in progress. By way of introducing technologically advanced training machinery into the rehabilitation programmes, the immediate therapeutic environment may well become far more conducive to effective work with the patients affected by FS.

This review is by no means free from certain limitations, however. Even though it is a systematic literature review as such, it has nevertheless been based on the studies published throughout the recent years, which focused predominantly on making use of modern technological solutions in working with older adults affected by FS. On the other hand, the present evaluation has not covered all modern methods in current use but concentrated on the ones the authors deemed both the most promising and boasting by far the greatest therapeutic potential in working with older adults.

## 5. Conclusions

To the best of the authors’ knowledge, well-grounded in their own hands-on rehabilitation experience with older adults affected by FS, it is now essential to rethink the traditional approach to rehabilitation and move on to applying the newly available solutions, as offered by VR technology, with a view to adapting conventional exercise regimen into something far more variable and attractive to the patients.

This narrative review indicates that modern technologies applied when working with the individuals affected by FS may well complement the traditional model of rehabilitation, or a comprehensive therapy, by enabling them to return from FS to the pre-frail stage whilst simultaneously enhancing both motor and cognitive function. Transforming the immediate training environment into the one far more conducive to regular training, with no need to make a trip to a traditional training facility, through online monitoring of the patient’s actual progress, and adequate execution of the exercise regimen in the pandemic and post-pandemic times, stands for a most essential factor in securing a desirable therapeutic outcome (we made an explicit reference to the COVID-19 pandemic as one of the reviewers asserted that ‘in the post-pandemic world, technology and telerehabilitation would become quite obvious’).

## Figures and Tables

**Table 1 ijerph-19-02985-t001:** Two main models of FS [1,4,9,10].

Phenotype Model	
**Unintentional weight loss**	>5 kg in 12 months
A sense of fatigue	Handgrip strength measurement, accounting for the subject’s age and body mass index (BMI)
Exhaustion	Depression scale (Center for Epidemiologic Studies Depression Scale, CESD)
Gait slowdown	Gait slowdown score ≥ 20 s in the ‘Stand up and walk’ test over a distance of 15 feet (approx. 4.6 m), accounting for the subject’s gender and height
Reduced physical activity	Shortened version of the Minnesota Leisure Time Activity Questionnaire (MLTAQ)
**Cumulative Deficit Model**	
Restriction of activity Anaemia and blood deficiency Arthritis Atrial fibrillation Cerebrovascular disease Chronic kidney disease Diabetes mellitus Dizziness Shortness of breath Falls Foot problems Broken bones Hearing impairment Heart failure Heart valve disease Inability to leave home Hypertension Hypotension/fainting	Ischaemic heart disease Memory and cognitive issues Mobility issues Osteoporosis Parkinsonism and tremors Peptic ulcer disease Peripheral vascular disease Polypharmacy Need for care Respiratory disease Skin ulceration Sleep disturbance Social vulnerability Thyroid disease Urinary incontinence Urinary tract disease Visual disturbances Weight loss and anorexia

**Table 2 ijerph-19-02985-t002:** The factors affecting incidence of FS in the developed and the developing countries [42,43,44].

**The developed countries** Age Female sex Black race Low level of education Body mass index Smoking Alcohol consumption Cardiovascular diseases Number of comorbidities Functional disability Poor self-assessment of health Depressive symptoms Poor cognitive functions Low income	**The developing countries** Age Female sex Low level of education Nutritional status Low physical activity Comorbidities Functional status Longest pursued occupation Low socioeconomic status

## Data Availability

The datasets generated and/or analysed during the current study are available from the Corresponding Author upon reasonable request.

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
