# Peer review of "Frailty Syndrome—Fall Risk and Rehabilitation Management Aided by Virtual Reality (VR) Technology Solutions: A Narrative Review of the Current Literature"

_ijerph, 2022, doi:10.3390/ijerph19052985_

Round 1
Reviewer 1 Report
This is very informative review about the use of technologies including virtual reality in rehabilitation of older persons. The article is clearly presented with the focus on the above mentioned. However, I would recommend to revise the English for easier reading. Very important it is especially in the abstract which is not clear and some sentences are difficult to understand.
Author Response
Authors' responses to the Reviewers
Title: Frailty Syndrome – Falls Risk and Rehabilitation Management aided by Virtual Reality (VR) Technology Solutions: A Narrative Review of Current Literature
Journal: International Journal of Environmental Research and Public Health
Special Issue:Frailty in the Elderly: Issues and Challenges
Submission ID:ijerph-1604732
The Authors have diligently addressed all the concerns raised by the Reviewers. Hopefully, the revised version of the manuscript will merit the Reviewers' satisfaction.
AUTHORS' RESPONSES TO THE REVIEWER'S RECOMMENDATIONS
REVIEWER # 1
|
|
|
Comments and Suggestions for Authors
This is very informative review about the use of technologies including virtual reality in rehabilitation of older persons. The article is clearly presented with the focus on the above mentioned. However, I would recommend to revise the English for easier reading. Very important it is especially in the abstract which is not clear and some sentences are difficult to understand.
AUTHORS' RESPONSE:
In line with the Reviewer's comments, the ABSTRACT has radically been re-edited and syntactically simplified to enhance its communicative appeal.

Reviewer 2 Report
This is generally a well written and timely paper that describes an emergent technology in a post-pandemic world in which technology and telerehabilitation will be quite evident. The review of the literature will be much appreciated by readers.
The paper loses some of its focus in reviewing adjunctive technologies and VR, sometimes discussing the two interchangeably. A more compelling paper would consider providing the more conceptual clarity equivalent to that the authors provide the reader on the concepts related to FS. This may cause some problem with word count, but the introduction could be shortened to get the reader to the main point (technology and VR) much quicker. It would also be helpful to the reader to "connect some of the dots" between the problems comprised by FS (well-discussed) and the rationale as to why technology and VR in particular might address those problems physiologically. It is absolutely true that technology in the home will have impact on access and costs. However, the authors should clearly identify why VR might impact falls therapeutically. Some space is given to other forms of technology such as sensors, but the emphasis of the paper, per the title, should be on VR.
Author Response
Authors' responses to the Reviewers
Title: Frailty Syndrome – Falls Risk and Rehabilitation Management aided by Virtual Reality (VR) Technology Solutions: A Narrative Review of Current Literature
Journal: International Journal of Environmental Research and Public Health
Special Issue:Frailty in the Elderly: Issues and Challenges
Submission ID:ijerph-1604732
The Authors have diligently addressed all the concerns raised by the Reviewers. Hopefully, the revised version of the manuscript will merit the Reviewers' satisfaction.
AUTHORS' RESPONSES TO THE REVIEWER'S RECOMMENDATIONS
REVIEWER # 2
Comments and Suggestions for Authors
This is generally a well written and timely paper that describes an emergent technology in a post-pandemic world in which technology and telerehabilitation will be quite evident. The review of the literature will be much appreciated by readers.
The paper loses some of its focus in reviewing adjunctive technologies and VR, sometimes discussing the two interchangeably. A more compelling paper would consider providing the more conceptual clarity equivalent to that the authors provide the reader on the concepts related to FS. This may cause some problem with word count, but the introduction could be shortened to get the reader to the main point (technology and VR) much quicker.
AUTHORS' RESPONSE:
The Authors are much indebted to the Reviewer for his pretty perceptive take of their narrative effort, and do actually concede his point on the conceptual clarity which might fall somewhat short of his expectations.
This said, the Authors have to pint out that should they opt to follow closely the actual recommendation, they would indeed encounter problems with remaining fully compliant with the strict word count constraints, let alone the complex logistics entailed in revamping the text effectively at a pretty short notice.
It's very much like taking bricks out of the wall, when trying to have it remodelled. Taking one out would require swapping the other bricks, and this would mean a more complex, and time-consuming effort.
It would also be helpful to the reader to "connect some of the dots" between the problems comprised by FS (well-discussed) and the rationale as to why technology and VR in particular might address those problems physiologically. It is absolutely true that technology in the home will have impact on access and costs. However, the authors should clearly identify why VR might impact falls therapeutically. Some space is given to other forms of technology such as sensors, but the emphasis of the paper, per the title, should be on VR.
AUTHORS' RESPONSE:
Once again, the Authors have to admit that the Reviewer's tip on "connecting some of the dots" hits the nail on the head, as this might well enhance overall focus on making use of VR technology, although the Authors would be faced with pretty much the same editing challenges, as already referenced further above.
Having said that, the Authors are quite happy to take the Reviewer's comments on board for future use, when putting together other review articles, which they might have in the pipeline shortly.

Reviewer 3 Report
The subject matter raised by the authors of the work is extremely important. In view of the aging of the population, especially in developed countries, it can be expected that the FS will concern an increasing percentage of citizens, and effective rehabilitation may contribute to the improvement of the length and quality of life.
However, I believe that some significant changes to the manuscript are needed which may further improve the quality and relevance of the text. In brackets, I give the numbers of lines to which the comment relates. I believe that there are many editorial and stylistic flaws in the text, which I have indicated below.
According to me, it would be valuable to mention the issue of the relationship between chronic lower limb ischemia and the frailty syndrome (you can use this recently published review paper describing it in details: doi.org/10.3390/ijerph17249339). Moreover, inflammatory process taking a part in pathogenesis of FS, participates also in pathogenesis of atherosclerotic cardiovascular disease as well as restenosis after its endovascular treatment, also in the context of peripheral arterial disease (doi.org/10.3390/ijerph182211970).
I don’t agree with the first sentence: “Frailty syndrome (FS) affects the older adults.” Although the frailty syndrome occurs more frequently in the elderly, it may occur also in younger persons. After definition of the frailty syndrome it should be mentioned that it affects mainly older people, but not only, but it should not be the first sentence of the paper because the age is not the most important aspect of FS. The most important aspect of FS is reduced capacity of physiological reserves and increased sensitivity to stressors affecting homeostasis of the body, and indeed it affects more frequently elderly. (44)
I’m not sure whether “physiological syndrome” is appropriate expression to the frailty syndrome. (45)
This paper is not a systematic review so in my opinion it is not appropriate to describing methodology of searching articles. (78-87) In my opinion, accurately describing the article search methodology in a regular review article (i.e. one that is not a systematic review) looks strange. Such a description of the methodology adds nothing here. In my opinion in this place the purpose of the study should be precisely described.
In my opinion the chapter “Pathophysiology” is chaotic and laconic. It would be worth to divide it into subsections, for example related to inflammatory process, nutritional aspects, endocrine dysregulation, muscle dysfunction, and others. An interesting recently published paper describing impact of diabetes to the development of FS is in this link: doi.org/10.3390/diabetology3010003 as well as doi.org/10.3390/endocrines2040047. Also genetic research give more and more interesting findings in the context of FS (for example doi.org/10.3390/genes13020231).
It should be not “1B”, but “1β”. (96)
It should be not “tumor necrosis factor alpha”, but “tumor necrosis factor α”. The rule of writing the Greek letters should be uniform. (97)
I don’t understand the abbreviation IP-10. Explain, please. (97)
It should be “a drop in concentration of ….” or “a drop in blood level of parameters such as….”. (98)
I think it should be “macro-”. (108)
I think it should be not “(≥ 30 ng/ml)”, but “not less than 30 ng/ml”. (124)
I don’t understand why 25(OH)D is written in brackets. (125)
I suggest to use not “this syndrome”, but „FS”. (49, 151)
I think, In English this expression should be written in capital letters (the Third World). (164)
It should be not „In the Control Group”, but „In the control group”. (247)
It should be not “Study Group”, but “study group”. (249, 251)
The chapters 4 and 5 are generally interesting and well written.
The chapter “Conclusions” is laconic, according to me and we can read that it is based mainly on Authors’ clinical experience, not on the results of review of the literature. Describe precisely the most important findings of your literature review, please.
The list of references is prepared in some positions not in accordance with the rules required in papers published by MDPI.
English seems to be quite good (the text is understandable for me, but I’m not English philologist and I feel not fully able to assess language quality).
Author Response
Authors' responses to the Reviewers
Title: Frailty Syndrome – Falls Risk and Rehabilitation Management aided by Virtual Reality (VR) Technology Solutions: A Narrative Review of Current Literature
Journal: International Journal of Environmental Research and Public Health
Special Issue:Frailty in the Elderly: Issues and Challenges
Submission ID:ijerph-1604732
The Authors have diligently addressed all the concerns raised by the Reviewers. Hopefully, the revised version of the manuscript will merit the Reviewers' satisfaction.
AUTHORS' RESPONSES TO THE REVIEWER'S RECOMMENDATIONS
REVIEWER # 3
Comments and Suggestions for Authors
The subject matter raised by the authors of the work is extremely important. In view of the aging of the population, especially in developed countries, it can be expected that the FS will concern an increasing percentage of citizens, and effective rehabilitation may contribute to the improvement of the length and quality of life.
However, I believe that some significant changes to the manuscript are needed which may further improve the quality and relevance of the text. In brackets, I give the numbers of lines to which the comment relates. I believe that there are many editorial and stylistic flaws in the text, which I have indicated below.
According to me, it would be valuable to mention the issue of the relationship between chronic lower limb ischemia and the frailty syndrome (you can use this recently published review paper describing it in details: doi.org/10.3390/ijerph17249339). Moreover, inflammatory process taking a part in pathogenesis of FS, participates also in pathogenesis of atherosclerotic cardiovascular disease as well as restenosis after its endovascular treatment, also in the context of peripheral arterial disease (doi.org/10.3390/ijerph182211970).
AUTHORS' RESPONSE:
All essential recommendations passed on by the Reviewer have been accommodated throughout the MS accordingly.
I don’t agree with the first sentence: “Frailty syndrome (FS) affects the older adults.” Although the frailty syndrome occurs more frequently in the elderly, it may occur also in younger persons. After definition of the frailty syndrome it should be mentioned that it affects mainly older people, but not only, but it should not be the first sentence of the paper because the age is not the most important aspect of FS. The most important aspect of FS is reduced capacity of physiological reserves and increased sensitivity to stressors affecting homeostasis of the body, and indeed it affects more frequently elderly. (44)
AUTHORS' RESPONSE:
In line with the Reviewer's perceptive observation, a reference to a much wider group of patients has been made instead of an original assertion. The opening sentence on FS has also been amended accordingly.
I’m not sure whether “physiological syndrome” is appropriate expression to the frailty syndrome. (45)
AUTHORS' RESPONSE:
We believe that ''physiological syndrome'' is the correct term for citing FS.
Let us quote the following in support of our assertion:
QUOTE:
''This study offers support for geriatricians' contention that frailty is a physiologic syndrome''
UNQUOTE
Excerpted from the study of Fried LP, Tangen CM, Walston J, Newman AB, Hirsch C, Gottdiener J, Seeman T, Tracy R, Kop WJ, Burke G, McBurnie MA; Cardiovascular Health Study Collaborative Research Group. Frailty in older adults: evidence for a phenotype. J Gerontol A Biol Sci Med Sci. 2001 Mar; 56(3): M146-56.
doi:10.1093/gerona/56.3.m146.
This paper is not a systematic review so in my opinion it is not appropriate to describing methodology of searching articles. (78-87) In my opinion, accurately describing the article search methodology in a regular review article (i.e. one that is not a systematic review) looks strange. Such a description of the methodology adds nothing here. In my opinion in this place the purpose of the study should be precisely described.
AUTHORS' RESPONSE:
In line with the Reviewer's recommendation, the passage on methodology has been replaced with the one referring to its aim.
In my opinion the chapter “Pathophysiology” is chaotic and laconic. It would be worth to divide it into subsections, for example related to inflammatory process, nutritional aspects, endocrine dysregulation, muscle dysfunction, and others. An interesting recently published paper describing impact of diabetes to the development of FS is in this link: doi.org/10.3390/diabetology3010003 as well as doi.org/10.3390/endocrines2040047. Also genetic research give more and more interesting findings in the context of FS (for example doi.org/10.3390/genes13020231).
AUTHORS' RESPONSE:
All essential recommendations passed on by the Reviewer have been accommodated in the MS accordingly.
It should be not “1B”, but “1β”. (96)
AUTHORS' RESPONSE:
Apologies for an obvious oversight, MS has been amended accordingly.
It should be not “tumor necrosis factor alpha”, but “tumor necrosis factor α”.
The rule of writing the Greek letters should be uniform. (97)
AUTHORS' RESPONSE:
Amended accordingly.
I don’t understand the abbreviation IP-10. Explain, please. (97)
AUTHORS' RESPONSE:
An inadvertent mental shortcut resulted in an oversight, rightly noted by the Reviewer. IP-10 stands for ,,interferon gamma-induced protein 10'', which has been amended accordingly.
It should be “a drop in concentration of ….” or “a drop in blood level of parameters such as….”. (98)
AUTHORS' RESPONSE:
In line with the Reviewer's comment, this concept has been clarified accordingly.
I think it should be “macro-”. (108)
AUTHORS' RESPONSE:
Amended accordingly.
I think it should be not “(≥ 30 ng/ml)”, but “not less than 30 ng/ml”. (124)
AUTHORS' RESPONSE:
Amended accordingly
I don’t understand why 25(OH)D is written in brackets. (125)
AUTHORS' RESPONSE:
Amended accordingly
I suggest to use not “this syndrome”, but „FS”. (49, 151)
AUTHORS' RESPONSE:
Amended accordingly
I think, in English this expression should be written in capital letters (the Third World). (164)
AUTHORS' RESPONSE:
Amended accordingly
It should be not „In the Control Group”, but „In the control group”. (247)
It should be not “Study Group”, but “study group”. (249, 251)
AUTHORS' RESPONSE:
Amended accordingly.
The chapters 4 and 5 are generally interesting and well written.
The chapter “Conclusions” is laconic, according to me and we can read that it is based mainly on Authors’ clinical experience, not on the results of review of the literature. Describe precisely the most important findings of your literature review, please.
AUTHORS' RESPONSE:
This passage has radically been re-edited to accommodate the Reviewer's criticism.
The list of references is prepared in some positions not in accordance with the rules required in papers published by MDPI.
AUTHORS' RESPONSE:
In line with the Reviewer's observation, the list of references has been reviewed again. Items 5, 51, 52 (italicised journal's name) and in ref. no 54 - deletion of number ,,6''.
English seems to be quite good (the text is understandable for me, but I’m not English philologist and I feel not fully able to assess language quality).

Reviewer 4 Report
The review is original and focuses on a target of people too often neglected
24-28 I suggest a synthesis as follow: “This narrative review aims to provide an overview of published studies focusing on therapeutic management approaches aided by virtual reality (VR) technology in frail elders at risk of falling.”
40: telerehabilitation, exergaming, elderly, seniors
46 There are many authors, who have provided different and often contradictory definitions of fragility for this I do not criticize the concept, but I recommend to include a bibliographic reference, perhaps even AuthorX et al. described frailty as …
53-58 I agree, too often we only think of musculoskeletal deficits related to frailty..
76 Intriguing why not make a figure instead of the table?
80-83 the string does not reflect what has been said in the title and abstract .. remove it as it is not a systematic context .. also because it is clumsily inserted between the FS section and the Fall risk
Sections 2 and 3 are massively useless and re-propose some concepts of the description of the frailty syndrome.
In this regard I recommend, to restructure the manuscript as follows ..
- Introduction (summary, background, rationale and objective narrative revision)
2. Fragility syndrome (little epidemiology, slight pathophysiology, clinical impact as already described in the introduction of this version of the paper)
3. Fall risk
4. VR
If it were possible, I would close the paragraph, with a summary sub-paragraph pre-conclusions with strengths and limitations. I think it would give readers a concise picture.
Author Response
Authors' responses to the Reviewers
Title: Frailty Syndrome – Falls Risk and Rehabilitation Management aided by Virtual Reality (VR) Technology Solutions: A Narrative Review of Current Literature
Journal: International Journal of Environmental Research and Public Health
Special Issue:Frailty in the Elderly: Issues and Challenges
Submission ID:ijerph-1604732
The Authors have diligently addressed all the concerns raised by the Reviewers. Hopefully, the revised version of the manuscript will merit the Reviewers' satisfaction.
AUTHORS' RESPONSES TO THE REVIEWER'S RECOMMENDATIONS
REVIEWER # 4
Comments and Suggestions for Authors
The review is original and focuses on a target of people too often neglected24-28 I suggest a synthesis as follows: “This narrative review aims to provide an overview of published studies focusing on therapeutic management approaches aided by virtual reality (VR) technology in frail elders at risk of falling.”
AUTHORS' RESPONSE:
In line with the Reviewer's observation, the ABSTRACT has been re-edited accordingly.
40: telerehabilitation, exergaming, elderly, seniors
AUTHORS' RESPONSE
AUTHORS' RESPONSES with regard to the missing response in Line 40: The MS has been amended accordingly to accommodate the Reviewer's comment.
46 There are many authors, who have provided different and often contradictory definitions of fragility for this I do not criticize the concept, but I recommend to include a bibliographic reference, perhaps even Author X et al. described frailty as …
AUTHORS' RESPONSE:
In line with the Reviewer's observation, the definition has been attributed accordingly.
53-58 I agree, too often we only think of musculoskeletal deficits related to frailty..
76 Intriguing why not make a figure instead of the table?
AUTHORS' RESPONSE:
Even though the Authors fully appreciate the Reviewer's point, they deliberately opted for a narrative clarification of the table's contents.
80-83 the string does not reflect what has been said in the title and abstract .. remove it as it is not a systematic context .. also because it is clumsily inserted between the FS section and the Fall risk
AUTHORS' RESPONSE:
In line with the Reviewer's recommendation, this passage has been deleted, and duly replaced with the one referring to the actual aims of the present narrative review.
Sections 2 and 3 are massively useless and re-propose some concepts of the description of the frailty syndrome.
In this regard I recommend, to restructure the manuscript as follows ..
- Introduction (summary, background, rationale and objective narrative revision)
2. Fragility syndrome (little epidemiology, slight pathophysiology, clinical impact as already described in the introduction of this version of the paper)
3. Fall risk
4. VR
AUTHORS' RESPONSE:
In line with the Reviewer's recommendation, the MS has been restructured accordingly.
If it were possible, I would close the paragraph, with a summary sub-paragraph pre-conclusions with strengths and limitations. I think it would give readers a concise picture.
AUTHORS' RESPONSE:
The closing paragraph has been altered in line with the Reviewer's recommendation.
The CONCLUSIONS section has been re-edited accordingly.

Round 2
Reviewer 1 Report
Thanks for amandmends. No further somments
Author Response
Authors' responses to the Reviewers
Title: Frailty Syndrome – Falls Risk and Rehabilitation Management aided by Virtual Reality (VR) Technology Solutions: A Narrative Review of Current Literature
Journal: International Journal of Environmental Research and Public Health
Special Issue:Frailty in the Elderly: Issues and Challenges
Submission ID:ijerph-1604732
The Authors have diligently addressed all the concerns raised by the Reviewers. Hopefully, the revised version of the manuscript will merit the Reviewers' satisfaction.
AUTHORS' RESPONSES TO THE REVIEWER'S RECOMMENDATIONS
REVIEWER # 1
Comments and Suggestions for Authors
The manuscript has been significantly improved. I recommend it for publication in the current form.
AUTHORS' RESPONSE:
The Authors are truly indebted to the Reviewer for his incisive comments and recommendations which tangibly contributed to enhancing its overall appeal.

Reviewer 2 Report
It is disappointing that the authors choose not to consider revisions with which they apparently agree.
Author Response
Authors' responses to the Reviewers
Title: Frailty Syndrome – Falls Risk and Rehabilitation Management aided by Virtual Reality (VR) Technology Solutions: A Narrative Review of Current Literature
Journal: International Journal of Environmental Research and Public Health
Special Issue:Frailty in the Elderly: Issues and Challenges
Submission ID:ijerph-1604732
The Authors have diligently addressed all the concerns raised by the Reviewers. Hopefully, the revised version of the manuscript will merit the Reviewers' satisfaction.
AUTHORS' RESPONSES TO THE REVIEWER'S RECOMMENDATIONS
REVIEWER # 2
Comments and Suggestions for Authors
It is disappointing that the authors choose not to consider revisions with which they apparently agree.
AUTHORS' RESPONSE:
The Authors very much regret that their original response has apparently been misconstrued by the Reviewer.
Having had to deal with four (4) reviewers at the same time proved a rather challenging experience, especially that each one of them had a different slant at the key issues, which eventually proved unfeasible to accommodate, so some sort of editorial consensus had to be reached, so as to retain overall coherence of the manuscript.
This said, the Reviewer might be pleased to notice (track-changes version of the MS) that substantial editing efforts, also structural, are now very much in evidence throughout the revised version of the manuscript.

Reviewer 3 Report
The manuscript has been significantly improved. I recommend it for publication in the current form.
Author Response
Authors' responses to the Reviewers
Title: Frailty Syndrome – Falls Risk and Rehabilitation Management aided by Virtual Reality (VR) Technology Solutions: A Narrative Review of Current Literature
Journal: International Journal of Environmental Research and Public Health
Special Issue:Frailty in the Elderly: Issues and Challenges
Submission ID:ijerph-1604732
The Authors have diligently addressed all the concerns raised by the Reviewers. Hopefully, the revised version of the manuscript will merit the Reviewers' satisfaction.
AUTHORS' RESPONSES TO THE REVIEWER'S RECOMMENDATIONS
REVIEWER # 3
Comments and Suggestions for Authors
The manuscript has been significantly improved. I recommend it for publication in the current form.
AUTHORS' RESPONSE:
The Authors are truly indebted to the Reviewer for his incisive comments and recommendations which tangibly contributed to enhancing its overall appeal.
Reviewer 4 Report
The manuscript has been enriched and restructured .. I can suggest to insert a figure that can accompany the pre-concluding section, also because it can increase the diffusion of the manuscript.
Unclear table 1 appears to be two different models of frailty, in fact it is a different diagnostic approach
In the fall risk section should be added to the risk assessment itself .. for example by referring to the stabilometric platforms and inertial sensors- ..
268 I suggest adding a description of the consoles as follows: “Relatively recent device technologies not created for rehabilitation approaches are assuming an intervention perspective for FS. Nintendo Wii is a commercial gaming device that uses a manual wireless controller and force platform. Xbox Kinect allows the user to interact with the game without a controller. Xbox Kinect does not provide visual references and space limits for sensors. The graphics environment of the Xbox Kinect is very detailed but less sensitive to detecting the displacement of the center of pressure than the Wii Balance Board. The latter, as a platform, on the other hand, allows participation in the training program even for patients who have less integrity in cognitive performance and can be considered a visual reference.” (ref: https://pubmed.ncbi.nlm.nih.gov/32478581/ )
Maybe I would mention the introduction of technologies such as BTS-NIRVANA ( ref: https://www.tandfonline.com/doi/abs/10.3109/09593985.2015.1138009 )
Limitations section before conclusion still missing.
Author Response
Authors' responses to the Reviewers
Title: Frailty Syndrome – Falls Risk and Rehabilitation Management aided by Virtual Reality (VR) Technology Solutions: A Narrative Review of Current Literature
Journal: International Journal of Environmental Research and Public Health
Special Issue:Frailty in the Elderly: Issues and Challenges
Submission ID:ijerph-1604732
The Authors have diligently addressed all the concerns raised by the Reviewers. Hopefully, the revised version of the manuscript will merit the Reviewers' satisfaction.
AUTHORS' RESPONSES TO THE REVIEWER'S RECOMMENDATIONS
REVIEWER # 4
Comments and Suggestions for Authors
The manuscript has been enriched and restructured .. I can suggest to insert a figure that can accompany the pre-concluding section, also because it can increase the diffusion of the manuscript.
Unclear table 1 appears to be two different models of frailty, in fact it is a different diagnostic approach
AUTHORS' RESPONSE:
The Authors have to admit they found their experience of having to deal with four (4) reviewers at the same time rather challenging, the more so that each one of them had a different slant. Consequently, not all recommendations could have been accommodated, as otherwise the manuscript's overall coherence would have suffered. The table at issue has therefore been retained by consensus.
In the fall risk section should be added to the risk assessment itself .. for example by referring to the stabilometric platforms and inertial sensors- ..
AUTHORS' RESPONSE:
Amended accordingly.
268 I suggest adding a description of the consoles as follows:
*“Relatively recent device technologies not created for rehabilitation approaches are assuming an intervention perspective for FS. Nintendo Wii is a commercial gaming device that uses a manual wireless controller and force platform. Xbox Kinect allows the user to interact with the game without a controller. Xbox Kinect does not provide visual references and space limits for sensors. The graphics environment of the Xbox Kinect is very detailed but less sensitive to detecting the displacement of the center of pressure than the Wii Balance Board. The latter, as a platform, on the other hand, allows participation in the training program even for patients who have less integrity in cognitive performance and can be considered a visual reference.”
(ref: https://pubmed.ncbi.nlm.nih.gov/32478581/)
AUTHORS' RESPONSE:
Amended accordingly.
The quote suggested by the Reviewer has been preceded by a brief intro clause, and duly supplemented with a pertinent reference.
Maybe I would mention the introduction of technologies such as BTS-NIRVANA
AUTHORS' RESPONSE:
Amended accordingly.
(ref:https://www.tandfonline.com/doi/abs/10.3109/09593985.2015.1138009)
Limitations section before conclusion still missing.
AUTHORS' RESPONSE:
Amended accordingly.
